# The Biological Fate of Pharmaceutical Excipient β-Cyclodextrin: Pharmacokinetics, Tissue Distribution, Excretion, and Metabolism of β-Cyclodextrin in Rats

**DOI:** 10.3390/molecules27031138

**Published:** 2022-02-08

**Authors:** Kunqian Mu, Kaiwen Jiang, Yue Wang, Zihan Zhao, Song Cang, Kaishun Bi, Qing Li, Ran Liu

**Affiliations:** 1School of Pharmacy, Shenyang Pharmaceutical University, 103 Wenhua Road, Shenyang 110016, China; szymkq@163.com (K.M.); kevinjiang0805@163.com (K.J.); 15940037209@139.com (Y.W.); zhao807991@163.com (Z.Z.); cangsong517@163.com (S.C.); bikaishun@126.com (K.B.); lqyxm@hotmail.com (Q.L.); 2School of Food and Drug, Shenzhen Polytechnic, 7098 Lau sin Avenue, Shenzhen 518115, China

**Keywords:** LC-MS/MS, β-cyclodextrin, pharmacokinetics, tissue distribution, excretion, metabolism

## Abstract

β-cyclodextrin has a unique annular hollow ultrastructure that allows encapsulation of various poorly water-soluble drugs in the resulting cavity, thereby increasing drug stability. As a bioactive molecule, the metabolism of β-cyclodextrin is mainly completed by the flora in the colon, which can interact with API. In this study, understanding the in vivo fate of β-cyclodextrin, a LC-MS/MS method was developed to facilitate simultaneous quantitative analysis of pharmaceutical excipient β-cyclodextrin and API dextromethorphan hydrobromide. The established method had been effectively used to study the pharmacokinetics, tissue distribution, excretion, and metabolism of β-cyclodextrin after oral administration in rats. Results showed that β-cyclodextrin was almost wholly removed from rat plasma within 36 h, and high concentrations of β-cyclodextrin distributed hastily to organs with increased blood flow velocities such as the spleen, liver, and kidney after administration. The excretion of intact β-cyclodextrin to urine and feces was lower than the administration dose. It can be speculated that β-cyclodextrin metabolized to maltodextrin, which was further metabolized, absorbed, and eventually discharged in the form of CO_2_ and H_2_O. Results proved that β-cyclodextrin, with relative low accumulation in the body, had good safety. The results will assist further study of the design and safety evaluation of adjuvant β-cyclodextrin and promote its clinical development.

## 1. Introduction

Pharmaceutical excipients are defined as any substance in drugs possessing pharmacological inertia, except for active pharmaceutical ingredients (API). The excipient added in the production of API formulations has various functions such as improving tablet performance, ensuring API dose uniformity, changing API release, and masking odor [1]. Excipients can influence the absorption, distribution, metabolism, and elimination process (ADME) of drugs administered jointly, and can also improve oral bioavailability by controlling drug dissolubility or penetrability [2]. The weight of excipients in any dosage form such as solid form, suspension, or solution is larger than that of active ingredients [3]. Excipients used to be considered inert, but in recent years, studies have predicted that excipients have a much more significant impact on the ADME process than drugs [4]. For a long time, researchers have paid great caution to the quality, function, and security of active components. Meanwhile, information about the study of excipients on dissolubility, penetrability, and their influence on absorption has been minimal. Until recently, studies have shown that it is necessary to check the quality and performance of excipients, as well as their safety in vivo [5,6]. In our previous studies on hydroxypropyl-β-cyclodextrin (HP-β-CD) in vivo, HP-β-CD was found to form a host–guest inclusion complex by non-covalent binding, affecting the metabolism of the API and increasing its AUC [7]. HP-β-CD is used for injection administration, but its analogue, β-cyclodextrin (β-CD), can only be used for oral administration; therefore, whether β-CD could have similar or opposing behavior in vivo relative to HP-β-CD, especially regarding its influence on API absorption, was the primary purpose of this study. 

β-CD is a cyclic oligosaccharide containing seven pyranose glucose units, which can be produced by amylose under the action of cyclodextrin glucosyltransferase [8]. β-CD has a conical three-dimensional texture with hydrophobic space and hydrophilic outer surface. Nevertheless, β-CD, as a unique excipient, has dual identities, both on changing API properties and efficacy. As an inclusion compound, β-CD can augment the dissolvability of hydrophobic drugs and change the pharmacokinetics [9]. As a bioactive molecule, β-CD can promote intracellular lipid accumulation and efflux and regulate intracellular lipid homeostasis, the metabolism of which is mainly completed by the flora in the colon. Therefore, either as an excipient or a potential effective substance, new possibilities for developing drugs with optimal pharmaceutical properties are afforded by the intensive study of the biological fate of β-CD in vivo. Presently, the literature states that high-performance liquid chromatography (HPLC) [10], evaporative light scattering detection (ELSD) [11], and pulsed amperometric detection (PAD) [12] have been used to quantify β-CD. These approaches achieved an LLOQ of at least 20 μg/mL, which is too high for trace amount detection in real utilization. The quality and physicochemical properties were mainly focused on in several researches of excipients, and the biological characteristics of β-CD in vivo have not received sufficient attention. 

The mass spectrometry detection method of HP-β-CD had been reported in our previous study [7]; however, due to the instability of the HP-β-CD molecule, it is decomposed into stable fragment ions at the first-stage quadrupole of mass spectrometry, so that the substance cannot be detected by MRM. Mass spectrometry detection of α-cyclodextrin had been reported in a previous study [13], but the SIM detection method is not suitable for the presence of more interference components, such as during metabolite analysis. The above methods were not applicable to the analysis of β-CD in vivo. Therefore, in this study, parent ion/product ion pairs scanning, with MRM in the positive mode, was applied to establish a sensitive and accurate method to analyze β-CD in vivo. In the process of seeking pharmaceutical ingredients, β-CD was discovered as the particular excipient in the commercially available Dextromethorphan Hydrobromide Granules (Guangdong Heping Pharmaceutical Co., Ltd., Guangdong, China), which could eradicate the disturbance of other excipients and focus solely on the study the biological fate of β-CD. At the same time, taking β-CD and dextromethorphan hydrobromide (DM) as examples, the interaction between medicinal excipients and active ingredients was researched further to gain insight into β-CD. In this research, first, an LC-MS/MS approach was established to simultaneous analyze pharmaceutical excipient β-CD and active pharmaceutical ingredient DM in rat plasma. It has been effectively used to study the pharmacokinetics, biological distribution, excretion, and metabolism of β-CD after oral medication of β-CD in rats. The coaction between β-CD and DM was soundly revealed by comparison of pharmacokinetics in vivo. A novel use of β-CD in pharmaceutical preparations is offered herein, further elucidating the fate of β-CD in vivo and progress of β-CD.

## 2. Materials and Methods

### 2.1. Chemicals and Reagents

β-cyclodextrin (β-CD) was acquired from Xiya Chemical Industry Co., Ltd. (Shandong, China). Dextromethorphan hydrobromide (DM), Ginsenoside Re (Internal Standard 1), and Psoralen (Internal Standard 2) were bought from Chengdu Chroma-Biotechnology Co., Ltd. (Chengdu, China). The structures of the aforementioned chemical ingredients are shown in Table 1. HPLC-grade acetonitrile, methanol, and formic acid were obtained from Fisher Chemicals (Fair Lawn, NJ, USA). Distilled water was obtained from Wahaha Group Co., Ltd. (Hangzhou, China).

### 2.2. Instruments and Conditions

#### 2.2.1. Quantitative Analysis Conditions

The liquid chromatographic separation process was operated on an XR LC-20AD Prominence^TM^ UFLC system (Shimadzu, Japan). At a column temperature at 40 °C, substances were analyzed on an Atlantis T3 (2.1 × 150 mm, 3 μm) column. The mobile phase system was water with 0.1% formic acid (A) and methanol with 0.1% formic acid (B). A flow rate of 0.4 mL/min and sample injection volume of 5 μL were used for gradient elution. The gradient elution program was established as follows: 0.01–0.3 min, 5% B; 0.3–0.8 min, 5–50% B; 0.8–1.5 min, 50–70% B; 1.5–2.8 min, 70–85% B; 2.8–5.0 min, 85% B; 5.0–5.01 min, 85–5% B; and 5.01–7.0 min, 5% B.

Mass spectrometry quantitative analysis was conducted on a Q TRAP 4000 MS/MS system to collect data, and then analyst software (version 1.6, AB Sciex, Framingham, MA, USA) was used to handle data. In the MRM mode and taking nitrogen as the circulating gas, electrospray positive ionization (ESI+) was used to analyze four substances. The detailed MS parameters were established as follows: gas 1, 50 psi; gas 2, 40 psi; air curtain gas, 40 psi; ion spray voltage, 5500 V; ion source temperature, 500 °C. Declustering potential (DP), entrance potential (EP), collision energy (CE), and cell exit potential (CXP) were performed as parameters optimized for substances in the MRM mode. 

#### 2.2.2. Qualitative Analysis Conditions

The metabolic samples were operated on an Agilent 1260 Infinity HPLC system. The gradient elution program was established as follows: 0.01–2.0 min, 5% B; 2.0–2.5 min, 5–50% B; 2.5–3.0 min, 50–70% B; 3.0–8.0 min, 70–75% B; 8.0–10.0 min, 75–85% B; 10.0–30.0 min, 85% B; 30.0–30.01 min, 85–5% B and 30.01–35.0 min, 5% B. Other liquid phase conditions were the same as those under Section 2.2.1.

Mass spectrometry qualitative analysis was conducted on a Q Triple TOF 5600 triple-time-of-flight hybrid mass spectrometer system with a DuoSpray^TM^ ion source to collect data. Then, the Metabolite Pilot software (version 1.5, AB Sciex, USA) was used to handle data. In the positive and negative mode, nitrogen was used as the circulating gas for analysis. The detailed MS parameters were established as follows: gas 1, 50 psi; gas 2, 50 psi; air curtain gas, 30 psi; ion spray voltage, 5500 V/−4500 V; ion source temperature, 550 °C; TOF MS scan, m/z 100–2000 Da; TOF MS DP, 80 V/−80 V; TOF MS CE, 10 V/−10 V; TOF MS/MS scan, m/z 50–1200 Da; TOF MS/MS DP, 80 V/−80 V; TOF MS/MS CE, 30 V/−30 V.

### 2.3. Preparation of Standard and Quality Control Samples

Accurately weighed β-CD, DM, and internal standards reference substances were prepared as stock solutions, respectively, with concentrations of 1000 μg/mL, 500 μg/mL, and 500 μg/mL with methanol. Then, the stock solutions were serially diluted with methanol to achieve standard working solutions of needed concentrations: DM was 0.25–1000 ng/mL and β-CD was 0.05–100 µg/mL. Standard solutions were prepared by adding the working solutions into blank rat samples; 10 µL of β-CD working solution was added into 100 µL of blank rat samples. In blank rat plasma, urine, and feces, β-CD was set at concentrations of 0.05, 0.2, 0.5, 2, 5, 7.5, 10 µg/mL as linear samples and 0.15, 2.5, 8.0 µg/mL as quality control (QC) samples. In tissue homogenate, β-CD was set at concentrations of 0.005, 0.02, 0.05, 0.2, 0.5, 2, 5 µg/mL as linear samples and 0.015, 0.25, 4.0 µg/mL as QC samples. In blank rat plasma, tissue homogenate urine and feces, DM was set at concentrations of 0.025, 0.1, 0.5, 2, 10, 50, 100 ng/mL as linear samples and 0.075, 2.5, 80 ng/mL as QC samples. All added internal standards concentration were 0.5 µg/mL. The lowest concentration of the linear sample was LLOQ. (The LLOQ of β-CD and DM were 0.05 µg/mL and 0.025 ng/mL, respectively.) All sample solutions were kept at 4 ℃ before applying.

### 2.4. Sample Preparation

#### 2.4.1. Sample Preparation for Quantification Analysis

The plasma and urine samples were kept at indoor temperature and dissolved before analysis. First, 100 µL of sample was added to a 1.5 mL Eppendorf tube together with 10 µL of ISs solution. Then, 300 µL of methanol:acetonitrile (1:1, *v*/*v*) was added, vortexed for 3 min, sonicated for 2 min in an ice-water bath, and centrifuged for 8 min at 16,000 rpm to deposit protein. The supernatant was transferred to a 1.5 mL Eppendorf tube, desiccated under airflow at room temperature, re-dissolved in 50 μL methanol, vortexed for 3 min, sonicated for 5 min in an ice-water bath, and centrifuged for 8 min at 16,000 rpm for LC-MS/MS analysis. For tissue and fecal samples, 100 mg of each sample was precisely weighed and added to a 5 mL Eppendorf tube, then mixed with 1 mL normal saline. The tissue cells were destructed by homogenization to mix with normal saline sufficiently, and the homogenization was achieved after centrifugation for 10 min at 4000 rpm. The methods to precipitate the protein of tissue and feces homogenization were the same as those of the plasma samples.

#### 2.4.2. Sample Preparation for Qualitative Analysis

Six rat plasma, tissue homogenates, feces, and urine samples were mixed evenly separately. First, 1 mL of sample was added to a 5 mL Eppendorf tube. Then, 3 mL of methanol:acetonitrile (1:1, *v*/*v*) was added, vortexed for 3 min, sonicated for 2 min in an ice-water bath, and centrifuged for 8 min at 16,000 rpm to deposit protein. The supernatant transferred to a 5 mL Eppendorf tube, desiccated under airflow at room temperature, re-dissolved in 80 μL methanol, vortexed for 3 min, sonicated for 5 min in an ice-water bath, and centrifuged for 8 min at 16,000 rpm for UHPLC-Q-TOF/MS analysis.

### 2.5. Animal Experiments

Male Wistar rats (250 ± 20 g) were obtained from Changsheng Biotechnology Co., Ltd. (Shenyan, China). The selected rats without specific-pathogen-free (SPF), lived in a SPF experimental animal environment, with temperature 18–26 °C and humidity 40–70%. Rats were given normal diet and drinking water. Rats were fasted for 12 h before blood collection in order to avoid some ingredients in food affecting the experimental results. All experimental routines followed the “Principles of Laboratory Animal Care”. The animal experiment was performed under the animal experiment guidelines of Shenyang Pharmaceutical University and approved by the Animal Ethics Committee. 

#### 2.5.1. Pharmacokinetics Study

Eighteen male Wistar rats were separated into three groups with six rats in every group based on different oral medicines, containing a β-CD-DM group (inclusion compounds of DM with β-CD), DM group (single DM), and β-CD group (single β-CD). Each rat was administered to orally, and the dosage of every group was computed by the equal transformation of human and rat surface area of the body by the clinical dosage [14]; therefore, 6 mg/kg of the β-CD-DM group (DM:β-CD(6 mg/kg:393 mg/kg) mixed evenly), 6 mg/kg of the DM group, and 393 mg/kg of the β-CD group were obtained. Blood samples (250 μL) were obtained from the orbital venous plexus at 0, 0.083, 0.25, 0.5, 0.75, 1, 1.5, 2, 4, 6, 8, 12, 24, 36 h after oral delivery. Then, the blood was centrifuged for 10 min at 4000 rpm to obtain the separation of plasma, and the plasma was stored in at −80 ℃ before application. The plasma samples of the rats in each group were treated according to the contents of Section 2.4.1, and then analyzed according to Section 2.2.1. The plasma concentrations of β-CD, DM, ginsenoside Re (IS 1), and psoralen (IS 2) were calculated by the ratio of peak area between analytes and internal standards. The pharmacokinetic parameters of the three groups of analytes were handled by adopting pharmacokinetic software DAS 2.1 found on non-compartment analysis. Then, the obtained main pharmacokinetic parameters of every group were used for independent sample T-test with SPSS v24.0; *p* < 0.05 was considered as a difference, and *p* < 0.01 was considered as a significant difference.

#### 2.5.2. Tissue Distribution Study

Eighteen male Wistar rats were separated into three groups with six rats in every group based on different times after administration, containing group A (30 min after oral administration), group B (1h after oral administration), and group C (4 h after oral administration). All Wistar rats were fed β-CD in a single oral administration at a dosage of 393 mg/kg. At 30 min, 1 h, and 4 h after oral administration, the rats were sacrificed by cervical dislocation. The rats were dissected immediately after death, and the tissues including heart, liver, spleen, lung, and kidney were collected. First, the tissues were sheared into a few pieces with scissors, and the rudimentary blood was washed with normal saline. Then, the tissues were stored at −80 ℃. The tissue homogenization samples of the rats in each group were treated according to the contents of Section 2.4.1, and then analyzed according to Section 2.2.1.

#### 2.5.3. Excretion Study

Six male Wistar rats (250 ± 20 g) were placed in metabolic cages one day before administration to adapt to the environment, and blank urine and fecal samples were collected. All Wistar rats were fed β-CD in a single oral administration at a dosage of 393 mg/kg, after which urine and feces were collected over the periods 0–12 h and 12–24 h. The urine and feces samples of the rats in each group were treated according to the contents in Section 2.4.1 and then analyzed according to Section 2.2.1.

#### 2.5.4. Metabolism Study

Six male SD rats were fed β-CD at a dosage of 393 mg/kg in continuous oral administration for 7d. Blood samples (250 μL) were obtained from the orbital venous plexus at 0, 0.25, 0.5, 1, 3, 6, 8, 12, 24 h after administration on the seventh day. Then, the blood was centrifuged for 10 min at 4000 rpm to obtain separation of plasma, and the plasma was kept at −80 ℃. Urine and feces were collected after administration. The rats were sacrificed by cervical dislocation 24 h after taking blood. The rats were dissected immediately after death and the tissues including heart, liver, spleen, lung and kidney were collected. The collected plasma, urine, and feces samples of rats in each group were treated according to the contents in Section 2.4.2, and the and then analyzed according to Section 2.2.2 The data gathered by UHPLC-Q-TOF-MS/MS were introduced into MetabolitePilot^TM^ software (version 1.5, AB Sciex, Framingham, MA, USA) for analysis. The minimum peak width and minimum retention time peak width were set to 25 ppm and 6 scans, respectively. The retention time and molecular weight error range were set to 0.50 min and 5 ppm, respectively.

### 2.6. Method Validation

The approach was conducted under the united instruction of the FDA Industrial Bioanalytical Method Validation guidelines and the Chinese Pharmacopoeia 2020 edition guidelines, including specificity, linearity, precision, accuracy, extraction recovery, matrix effect, stability, and dilution reliability.

Specificity was evaluated by the differences of typical chromatograms of blank samples, blank samples added with analyte, and actual samples after oral administration.

Two calibration curves were constructed daily and analyzed for three consecutive days to evaluate linearity.

Six replicates of LLOQ and six replicates of three concentrations of QC samples were prepared and analyzed for three consecutive days to assess intra-day and inter-day precision and accuracy.

The recovery rate was determined by comparing the peak area ratio of analyte and IS in six repeats of three concentrations of QC samples. 

The internal standard normalized matrix effect factor (IS-normalized MF) was used to evaluate the matrix effect.

The low and high QC samples were stored at room temperature for 7 h, in the autosampler for 8 h, low temperature for a long time (−80 °C for one month), after three freeze–thaw cycles. Then, the analytes were analyzed to evaluate the stability.

Dilution reliability was calculated by analyzing six repeats of 80 μg/mL samples (10 folds the maximum QC concentration) diluted to 8.0 μg/mL with rat urine and feces.

## 3. Results and Discussion

### 3.1. Method Development

This approach aimed to achieve sensitive and accurate detection of the β-CD in rat plasma. Because of the properties of cyclic oligosaccharides, the structure of the open chain was occupied less often, and it principally resided as an invariable circle structure in the process of ionization [15]. In this research, positive ion mode had a better signal than negative ion mode. Consequently, we measured the substances in the positive ion mode to get more sensitive experimental results. The quantitative parameters are listed in Table 1. Full-scan product ion spectrums are listed in Figure 1. To simultaneously determining the contents of β-CD and DM in rat plasma, the extraction results of the analytes with different solvents and the experimental results of pretreatment of DM and β-CD samples by liquid–liquid extraction and protein precipitation were compared in this study. The recovery rate of other solvents showed that when methanol:acetonitrile (1:1, *v*/*v*) was chosen as an extraction agent for DM and β-CD, the extraction recovery rate was 90%, the matrix effect was good, the specificity was strong, and the sensitivity was high. In the majority of liquid phase terms, the chromatographic column played an important role in the separation of various analytes. It had strong polarity and weak retention because of the polyhydroxy cyclic structure of β-CD. By contrast, DM possessed a feebleness polarity feature. On account of the enormous polarity differences between β-CD and DM, a chromatographic column that provides satisfactory retention nature for both strong and feebleness polarity substances was selected for simultaneously detecting the two compounds. Atlantis T3, which is a highly hydrophilic reversed-phase chromatography column, could be a reliable selection. Two analytes and internal standards were isolated in 7 min by uncomplicated gradient elution on the Atlantis T3 column with greater separation. Water with 0.1% formic acid (A) and methanol with 0.1% formic acid (B) were chosen as the mobile phases because of the resulting satisfactory chromatogram shapes and better separation effect. Ginsenoside Re (IS 1) was chosen as IS of β-CD, and psoralen (IS 2) was chosen as IS of DM due to its appropriate retention time and mass response compere to analytes.

### 3.2. Method Validation

#### 3.2.1. Specificity

The retention times of the typical chromatograms of blank plasma, blank plasma spiked with β-CD, DM, ISs, and plasma samples after oral medication were assessed to determine the specificity of β-CD and DM. There was no obvious peak disturbance at the retention times of β-CD (2.40 min) and DM (3.28 min), and Ginsenoside Re (IS 1) (3.61 min) and psoralen (IS 2) (3.70 min), as shown in Figure 2, cueing that no intrinsic compounds influenced the separation of β-CD and DM. 

#### 3.2.2. Linearity

There was a good linearity from 0.05 to 10 µg/mL of β-CD (R^2^ = 0.9977) and from 0.025 to 100 ng/mL of DM (R^2^ = 0.9970). The linearity of calibration curves of β-CD and DM is shown in Table 2. The results showed that the regression coefficients of the analytes were greater than 0.99, indicating a good linear relationship.

#### 3.2.3. Precision and Accuracy

In order to assess the precision (RSD) and accuracy (RE) of the method, the RSD and RE for LLOQ and QCs at three concentration levels, repeated six times, were determined. The inter-day RSD ranged from 6.10–12.96%, and the intra-day RSD ranged from 2.12–7.76%. The RE ranged from 0.02–6.28%. The detailed accuracy and precision of β-CD and DM are shown in Table 3, suggesting that our method could accurately and repeatedly quantify β-CD and DM in rat plasma.

#### 3.2.4. Extraction Recovery and Matrix Effect

The extraction recovery of β-CD was greater than 81.22%, that of DM was greater than 85.51%, and that of the internal standards was greater than 91.01%, verifying the effectiveness of the extraction method quantitatively. The matrix effect of β-CD was 91.21–92.80% and DM was 88.74–114.20%, indicating that the endogenous matrix during sample processing did not affect the quantification of the analytes.

#### 3.2.5. Stability

The stability RSD of β-CD and DM was 0.51–5.53%. The stability results showed that the samples could still be accurately quantified after being stored under different conditions.

#### 3.2.6. Dilution Reliability

Note that due to the high concentration of β-CD in rat urine and feces, which exceeded the set range of the standard curve, we need to ensure dilution reliability. The RE and RSD of dilute QC samples were within 15%, verifying that the dilution process had no effect on the quantification of β-CD. The higher concentration of samples that exceeded the range of linearity could be properly diluted to precisely measure the concentration of β-CD. 

To ensure the accuracy and applicability of tissue distribution research and excretion research results, rat blank mixed tissue homogenization (heart, liver, spleen, lung, and kidney blank tissue homogenization samples), rat blank urine and feces homogenization were used as the matrix to method validation, similar to rat plasma. 

The above results demonstrated that our approach could exactly and repeatedly measure the concentration of β-CD and DM in rat samples.

### 3.3. Pharmacokinetics Study

The average plasma concentrations over time in rats after single oral administration of β-CD and DM are shown in Figure 3. The main pharmacokinetic parameters are shown in Table 4. 

The differences between inclusion compounds and single compounds were mainly reflected in the pharmacokinetic parameters AUC_0−∞_, Cmax, t1/2, and CLZ. From Figure 3A, it was observed that after the oral administration of the exact dosage of β-CD, regardless of whether the excipients contain the active pharmaceutical ingredients, the pharmacokinetic behavior of β-CD in rats had the analogous trend: the Cmax of inclusion compounds was apparently lower than that of single compounds. However, from Figure 3B, the plasma concentration of DM after oral delivery was significantly different in two groups. It can be seen from the change in guest DM that compared with a single-compound group, AUC_0−∞_ and t1/2 of DM in inclusion compounds were significantly increased, and CLZ was decreased.

From the concentration–time plots of β-CD in Figure 3A, it was discovered that compared with the single-compound groups, the curve of the inclusion compounds group was smoother, the elimination was slower, AUC, t1/2 in the inclusion compounds group of β-CD had no significant difference. Moreover, the Cmax of β-CD of the inclusion compounds group was obviously decreased. The reason for this might be the dynamic balance between the host and guest and the inclusion compounds. The hydroxyl group of β-CD was outward, which made it highly hydrophilic. In addition, due to the role of the glycosidic oxygen bond, the hydrophilic area inside the cavity was poor. The conical three-dimensional structure of β-CD with hydrophobic space and hydrophilic outer surface made the β-CD (host) incorporate other drugs (guest) into the cavity to form a so-called host–guest inclusion compound [16]. As for the host molecule β-CD, space in the lipophilic central area was superseded by the lipophilic part of the guest molecule DM, which degraded the utilization of β-CD in aggregates and led to descent in Cmax [7,17]. In this study, AUC_0–t_ accounted for more than 96% of AUC_0–∞_, demonstrating that β-CD was completely eliminated in rat plasma 36 h after administration.

The guest molecule DM is poorly water-soluble and belongs to the low-solubility and high-permeability drugs of the II class in the biopharmaceutics classification system (BCS), and its dissolution is a rate-limiting process, so it is mainly sold in the form of its hydrobromide (HBr) salt form [18]. In general, oral absorption of BCS II drugs in biopharmaceutical classification systems is enhanced by β-CD [19]. It was found from the plasma concentration–time curve of DM in Figure 3B that compared with the single compounds group, DM in the inclusion compounds group showed better bioavailability of DM. Once the DM guest drug molecules entered the β-CD hydrophobic space, conformational adjustment would be undergone to maximize the use of weak van der Waals forces [20]. It had been reported that hydrogen bonds played an essential role in the aggregation of β-CD inclusion compounds. The -OH group of β-CD could form hydrogen bonds between guest drugs and β-CD for connection [21]. The existence of hydrogen bonds could enhance the solubility of DM in plasma [22] and promote the release of DM from the hydrophobic space more [23]. Moreover, the stability of the DM in body liquid could also be improved, so that the exposure amount of DM in rats was more significant than the single compounds group [24], resulting in a significant increase in AUC_0–∞_. There was covalent bond formation or breaking during the construction of the inclusion compounds. It took longer for guest and host molecules to be in a dynamic balance, with t_1/2_ significantly increased and CL_Z_ significantly decreased [25].

### 3.4. Tissue Distribution Study

The result of tissue distribution are shown in Figure 4A. After oral administration for 30 min, β-CD was detected in all tissues, displaying that β-CD was rapidly and widely distributed. In the early distribution period, β-CD was constantly distributed in organs with a richness of blood flow, for instance, kidney and liver with the highest concentration. At this time, the concentration of β-CD in rat plasma (~3.5 μg/mL) indicated that β-CD did not achieve a dynamic balance among plasma and tissues. At 1 h, the concentration of β-CD in entire tissues reached its peak, and it was higher in the spleen, liver, and kidney, because these organs have high perfusion rates. Because cyclodextrin can enter macrophages, which are widely distributed in the liver and spleen [26], macrophages will swallow a lot of β-CD, and therefore the concentration of it in the liver and spleen would be higher. In addition, although the heart is an organ with a high perfusion rate, the concentration of β-CD in the heart was significantly lower than that in the spleen and liver with a high perfusion rate, which might be due to the lower distribution of phagocytes in the heart [27]. After oral administration of β-CD, the concentration of β-CD in the kidney was significantly higher than other organs, indicating that β-CD has a trend of renal enrichment, mainly through renal excretion. This may be because of the low water solubility of β-CD, which may cause renal damage and hemolysis [28]. At this time, the concentration of β-CD in tissues was similar to plasma (~3.0 μg/mL), representing that the distribution of β-CD in tissues and plasma was approaching dynamic balance. At 4 h, the drug concentration in the tissues decreased significantly without noticeable accumulation. This study proved that β-CD would not cause significant accumulation of drugs in vivo, and further demonstrated that β-CD was safe as a medicinal excipient.

### 3.5. Excretion Study

The excretion results are shown in Figure 4B. A high concentration of β-CD was detected in feces. It was reported that β-CD was poorly absorbed and rapidly excreted in the gastrointestinal tract [29]. When β-CD was administered orally, its absorption was extraordinarily low, and it was mostly excreted into feces [30]. Cyclodextrins could be excreted through kidneys and rapidly excretion through urine [31]. The content of β-CD in urine was lower than that in feces, possibly due to its larger size: as size increases, kidney excretion decreases [26]. Whole-body absorption of β-CD was low after oral administration. After oral administration for 1–12 h, the cumulative excretion of intact β-CD to urine and feces was 108.5 μg/mL, which was lower than the administration dose (393 mg/kg). It showed that oral cyclodextrin was mostly metabolized in the intestine, and a small part was metabolized through the kidney. The concentration of β-CD in urine and feces for 12–24 h was low. Over time the concentration of β-CD in plasma and tissues gradually decreased, and a large part was discharged in vitro through the intestinal tract and kidney metabolism. The results showed that β-CD was eliminated rapidly in vivo.

### 3.6. Metabolism Study

Cyclodextrin was considered as the carrier of active ingredients until paying attention to the possible complexation of essential components in the intestine or blood, and it is speculated that it has physiological effects and participates in the metabolism in vivo [16]. Because of the low excretion of β-CD through urine and feces after oral administration and the limited systemic absorption of β-CD, it could predicted that β-CD was involved in the metabolic process in vivo, which might be eliminated mainly through the metabolic process.

In fact, β-CD has resistance to gastric acid or saliva and pancreatic amylase, and it is widely hydrolyzed in the colon. β-CD is a cyclic oligosaccharide with non-reducing properties consisting of glucose units connected by α-1,4-glycosidic bonds [32], and a ring-opening reaction may occur in primary metabolites and acyclic maltodextrin, becoming linked by an α-1,4-glycosidic bond [33]. The metabolic results showed that di-oxidation and fetone formation occurred in kidney and plasma samples, and I metabolism occurred. Di-oxidation reactions can occur in any branched chain, in this case, R-CH_2_-R_1_ transforming into R-CO-R_1_, with a carboxyl (-COOH) group forming at the branched chain. In fecal samples, the oxidation reaction of β-CD in the branched-chain formed carboxyl (-COOH) as I metabolism, followed by taurine conjugation reaction as per II metabolism, which was R-COOH transforming into R-CONH-CH_2_CH_2_SO_3_H. Maltose amylase was generated in the cytoplasm and displayed higher hydrolytic competence to β-CD, and the products were mainly maltose and sugar molecules of various lengths [34]. These products were further metabolized, absorbed, and eventually discharged in the form of CO_2_ and H_2_O [30]. Moreover, the results showed that the metabolites produced had no toxic reaction.

## 4. Conclusions

In this research, a specific and accurate LC-MS/MS approach was developed for the quantitative measurement of dextromethorphan hydrobromide and β-CD. Simultaneous determination and pharmacokinetic study of active ingredients and excipient in pharmaceutical preparations were achieved first. To offer an integrative representation of the biological fate of β-CD, the pharmacokinetics, tissue distribution, excretion, and metabolism of excipient β-CD were studied. The results showed that the effect of inclusion compounds changed the biological process, elimination rate, and exposure amount of β-CD and dextromethorphan hydrobromide in vivo. The study showed that β-CD was almost wholly removed from rat plasma within 36 h, and high concentrations of β-CD existed in organs with a high perfusion rate. A large proportion of β-CD was excluded from urine and feces, and it might be metabolized to maltodextrin. Sugar molecules with various lengths were further metabolized, absorbed, and finally expelled in the form of CO_2_ and H_2_O. Our results showed that β-CD had relative low accumulation in the body and the excretion and metabolism of β-CD were relatively rapid in vivo. Our results will be helpful to further study the design and safety evaluation of adjuvant β-CD and promote its clinical development.

## Figures and Tables

**Figure 1 molecules-27-01138-f001:**
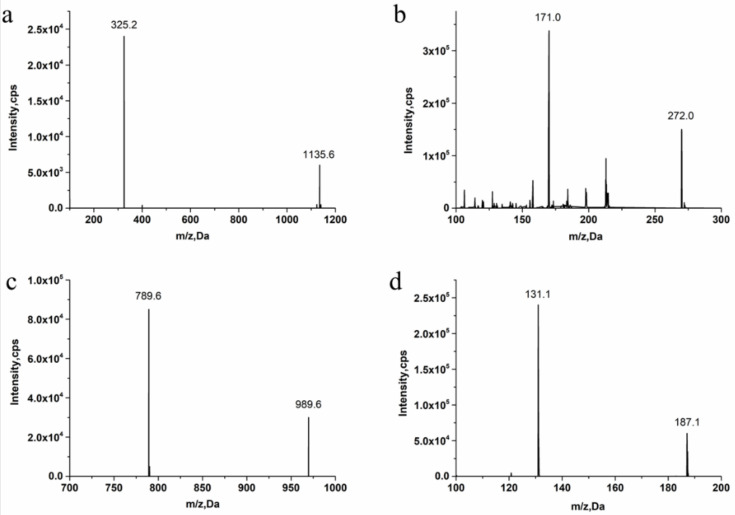
Full-scan product ion spectrums of β-cyclodextrin (β-CD) (**a**) (*m*/*z* 1135.6/325.2), dextromethorphan hydrobromide (DM) (**b**) (*m*/*z* 272.0/171.0), ginsenoside Re (IS1) (**c**) (*m*/*z* 989.6/789.6), and Psoralen (IS2) (**d**) (*m*/*z* 187.1/131.1).

**Figure 2 molecules-27-01138-f002:**
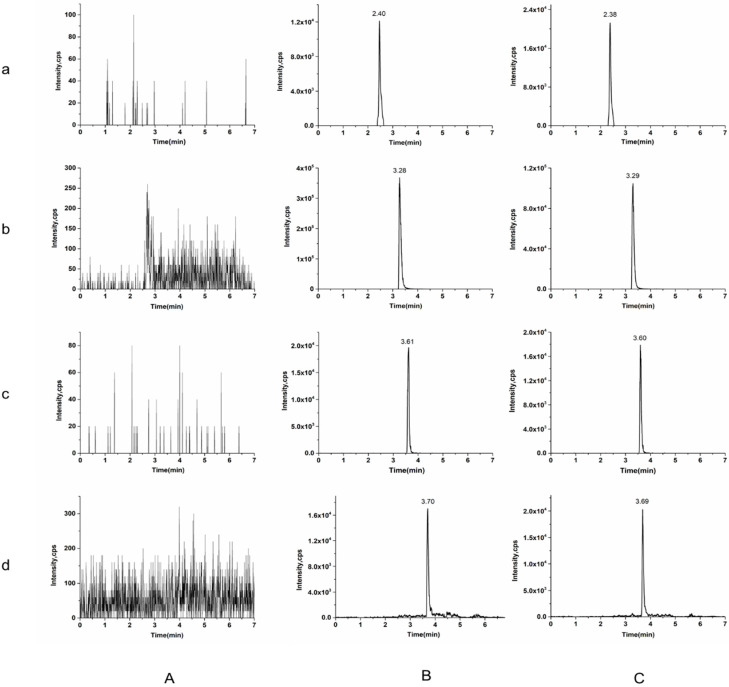
Chromatograms for β-CD (**a**), DM (**b**), ginsenoside Re (IS 1) (**c**), and Psoralen (IS 2) (**d**) in a blank plasma sample (**A**), a blank plasma spiked with 5 μg/mL of β-CD, 50 ng/mL of DM and 0.05 μg/mL of ISs (**B**) and a plasma sample 1 h after oral administration (**C**).

**Figure 3 molecules-27-01138-f003:**
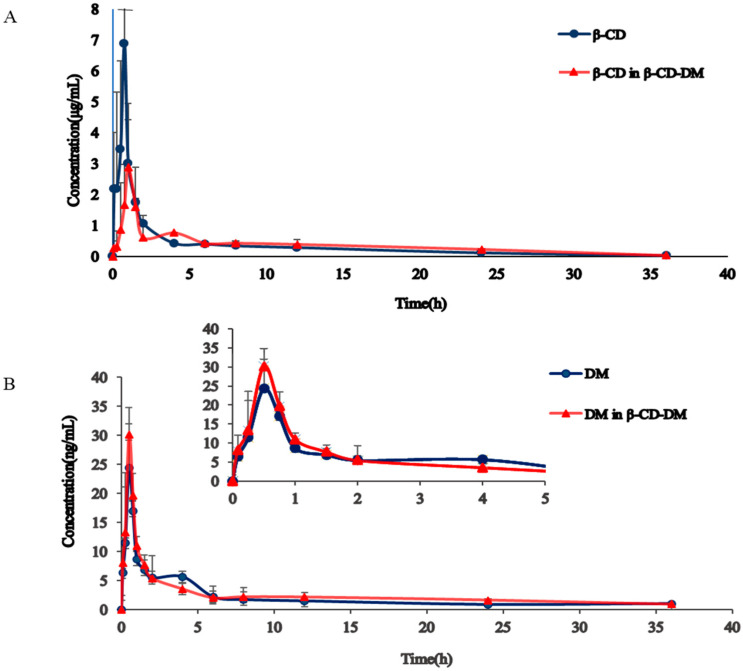
Plasma concentration–time plots of β-CD (**A**) and DM (**B**) after oral delivery in two groups.

**Figure 4 molecules-27-01138-f004:**
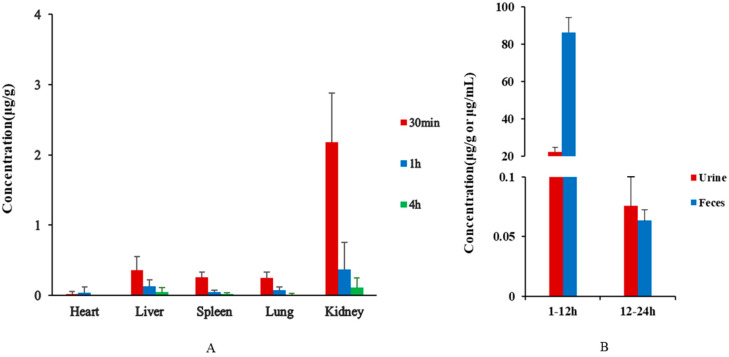
Concentrations of β-CD in rat tissues at different time points (**A**) and concentrations of β-CD in rat urine and feces at different time periods (**B**) after oral administration.

**Table 1 molecules-27-01138-t001:** The structures and quantitative parameters of β-CD, DM, ginsenoside Re (IS1), and Psoralen (IS2).

Analyte	Structure	Q1Mass (Da)	Q3Mass (Da)	DP (V)	EP (V)	CE (V)	CXP (V)
β-CD	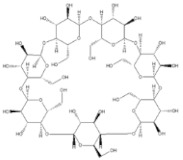	1135.6	325.2	142.7	8.4	44.8	16.4
DM	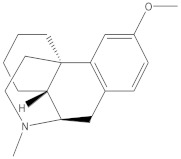	272.0	171.0	94.1	7.3	52.5	11.0
ginsenoside Re (IS1)	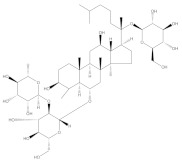	969.6	789.6	207.1	10	61.7	13
Psoralen (IS2)	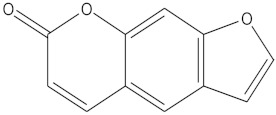	187.1	131.1	80	11	35	10

**Table 2 molecules-27-01138-t002:** The linearity of the calibration curves of β-CD and DM concentration in rat samples.

Sample	Analyte	Liner Range (μg/mL)	Slope	Intercept	Regression Coefficient
Plasma	β-CD ^a^	5.0000 × 10^−2^–1.0000 × 10^1^	4.1848 × 10^−2^	1.6829 × 10^−1^	0.9977
	DM ^b^	2.5000 × 10^−5^–1.0000 × 10^−1^	1.6445 × 10^−2^	3.4248 × 10^−2^	0.9970
Tissue	β-CD ^a^	5.0000 × 10^−3^–5.0000	5.8375 × 10^−2^	1.2174 × 10^−2^	0.9952
Urine	β-CD ^a^	5.0000 × 10^−2^–1.0000 × 10^1^	6.0808 × 10^−2^	2.6697 × 10^−1^	0.9991
Feces	β-CD ^a^	5.0000 × 10^−2^–1.0000 × 10^1^	5.7079 × 10^−2^	3.0097 × 10^−1^	0.9971

^a^: Used ginsenoside Re (IS1) as the internal standard of β-CD; ^b^: Used Psoralen (IS2) as the internal standard of DM.

**Table 3 molecules-27-01138-t003:** Accuracy and precision of β-CD and DM in rat plasma (mean ± SD, n = 6).

Sample	Analyte	Concentration	Concentration (μg/mL)	Inter-Day (RSD,%)	Intra-Day (RSD,%)	Accuracy (RE,%)
Plasma	β-CD	LLOQ	0.05	12.96	7.76	−1.70
		QC1	0.15	9.70	6.66	−0.02
		QC2	2.50	9.18	2.51	3.77
		QC3	8.00	6.10	6.18	3.82
	DM	LLOQ	2.50 × 10^−5^	11.60	4.53	-1.50
		QC1	7.50 × 10^−5^	7.11	7.11	−3.30
		QC2	2.5 × 10^−3^	9.60	2.12	6.28
		QC3	8.0 × 10^−2^	11.19	6.10	−2.42
Tissue	β−CD	LLOQ	0.05	7.12	4.33	−0.32
		QC1	0.15	13.62	4.72	−0.26
		QC2	2.50	2.73	2.53	0.70
		QC3	8.00	8.28	3.03	−1.22
Urine	β−CD	LLOQ	0.05	14.40	6.70	−6.86
		QC1	0.15	6.97	4.22	−2.55
		QC2	2.50	9.70	7.07	0.71
		QC3	8.00	11.78	4.13	0.67
Feces	β−CD	LLOQ	0.05	5.34	7.76	−1.79
		QC1	0.15	5.66	5.26	−5.23
		QC2	2.50	4.63	6.35	5.24
		QC3	8.00	14.55	2.74	6.89

**Table 4 molecules-27-01138-t004:** Pharmacokinetic parameters of β-CD and DM in three groups (mean ± SD, n = 6).

Analyte	Groups	AUC_0–t_ (ng/mL/h)	AUC_0–∞_ (ng/mL/h/)	C_max_(ng/mL/h)	t_1/2_(h)	CL_Z_ (L/h/Kg)	V_Z_ (L/Kg)
β-CD	β-CD	1.33 × 10^4^ ± 6.51 × 10^3^	1.38 × 10^4^ ± 6.38 × 10^3^	7.30 × 10^3^ ± 2.31 × 10^3^	8.33 ± 2.46	3.27 × 10^1^ ± 1.02 × 10^1^	4.00 × 10^2^ ± 1.76 × 10^2^
	β-CD-DM	1.32 × 10^4^ ± 2.80 × 10^3^	1.37 × 10^4^ ± 2.90 × 10^3^	3.81 × 10^3^ ± 6.99 × 10^2^ **	8.21 ± 1.47	3.03 × 10^1^ ± 6.18	3.54 × 10^2^ ± 7.16 × 10^1^
DM	DM	7.73 × 10^1^ ± 1.64 × 10^1^	9.28 × 10^1^ ± 1.11 × 10^1^	2.91 × 10^1^ ± 3.17	1.68 × 10^1^ ± 8.56	6.55 × 10^1^ ± 8.60	1.64 × 10^3^ ± 9.96 × 10^2^
	β-CD-DM	9.22 × 10^1^ ± 3.14 × 10^1^	1.47 × 10^2^ ± 4.93 × 10^1^ *	3.27 × 10^1^ ± 4.67	3.04 × 10^1^ ± 7.48 *	4.41 × 10^1^ ± 1.25 × 10^1^ **	1.90 × 10^3^ ± 6.43 × 10^2^

AUC_0–t_, area under the concentration–time curve from time zero to the time of the last measurable concentration; AUC_0−∞_, area under the concentration–time curve from time zero to infinity; Cmax, peak plasma concentration; t1/2, half-time; CLz, clearance rate; Vz, apparent volume of distribution. * *p* < 0.05 compared with the normal group. ** *p* < 0.01 compared with the normal group.

## Data Availability

Not applicable.

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
