# Peer review of "The Biological Fate of Pharmaceutical Excipient β-Cyclodextrin: Pharmacokinetics, Tissue Distribution, Excretion, and Metabolism of β-Cyclodextrin in Rats"

_molecules, 2022, doi:10.3390/molecules27031138_

Round 1
Reviewer 1 Report
R 180: I consider that a more detailed description of the microclimate and feeding conditions of the rats should be described, as these may directly influence the course of the experiment.
R483: You state that: It proved that β-CD has good safety. From my point of view, it is not the best expression because I have not performed toxicological studies to demonstrate safety, e.g. lethal dose 50. In conclusion, I recommend reformulating this sentence.
This study brings tangible benefits in the study of cyclodextrins by demonstrating the effectiveness of this method of monitoring their course in vivo and the possibility of predicting the amount of load of the active substance in cyclodextrin reach the percentage in organs studied per unit time. This may lead to the thinking of new therapeutic strategies that, on the one hand, increase the solubility of the active substance incorporated in β-CD, and on the other hand, address these cyclodextrins as possible vectors for specific organs depending on the unit of time.
Reviewer 2 Report
The research manuscript describes about evaluating the biological fate of pharmaceutical excipient β-cyclodextrin in rats and studying its pharmacokinetics, tissue distribution, excretion, metabolism in presence and absence of API. The conclusion is drawn from ample amount of validated data however there are following comments which can help to improve the manuscript before considering for publication.
Comments:
- Page 2, line 61, to give reference to the following line ‘As an inclusion compounds, β-CD can augment the dissolvability of hydrophobic drugs and change the pharmacokinetics’, following reference may be cited which describes about use of beta cyclodextrin in presence of hydrophobic drug in liposome and evaluating its effects on membrane integrity and dissolvability of liposomes. Mashru R et al. Liposomes encapsulating native and cyclodextrin enclosed Paclitaxel: Enhanced loading efficiency and its pharmacokinetic evaluation, International Journal of Pharmaceutics. (2018); 536, 95-107.
- For animal study, why only male rats were evaluated? How authors can justify fate in female rats?
- Authors have mentioned that 6 animals per group were used. How the number of animals were decided? Was there any power analysis run to find out the number of animals needed per group?.
- How the dose 6 mg/Kg was decided. Please give reference
